

**Factors controlling *Carex brevicuspis* leaf litter**
**decomposition and its contribution to surface soil organic**
**carbon pool at different water levels**
Lianlian Zhu[1,2,3], Zhengmiao Deng[1,2,*], Yonghong Xie[1,2, *], Xu Li[1,2], Feng Li[1,2],
Xinsheng Chen[1,2], Yeai Zou[1,2], Chengyi Zhang[4], Wei Wang[1,2]
[1] Key Laboratory of Agro-ecological Processes in Subtropical Region, The Chinese
Academy of Sciences, Changsha 410125, China.
[2] Dongting Lake Station for Wetland Ecosystem Research, Institute of Subtropical
Agriculture, The Chinese Academy of Sciences, Changsha 410125, China.
[3] University of Chinese Academy of Sciences, Beijing 100049, China.
[4] National Climate Center, China Meteorological Administration, Beijing 100081,
China.
*\* Correspondence*:
Corresponding Author:
Zhengmiao Deng: dengzhengmiao@163.com
Yonghong Xie: yonghongxie@163.com





**Abstract.** Litter decomposition plays a vital role in wetland carbon cycling. However, the contribution of aboveground litter decomposition to the wetland soil organic carbon (SOC) pool has not yet been quantified. Here, we conducted a *Carex brevicuspis* leaf litter input experiment to clarify the intrinsic factors controlling litter decomposition and quantify it's contribution to SOC pool at different water levels. This species is ubiquitous to global freshwater wetlands. We sampled this plant leaf litter at -25, 0, and +25 cm relative to the soil surface over 280 days and analysed leaf litter decomposition and its contribution to the SOC pool. The mass loss and carbon release rates were the highest at +25 cm water level, followed by the 0 cm water level. The rates of these parameters were the lowest at -25 cm water level. Significant amounts of litter carbon, nitrogen, and phosphorus were released at all three water levels. Litter input significantly increased the soil microbial biomass and fungal density but had nonsignificant impacts on soil bacteria, actinomycetes, and fungal/bacterial concentrations at all three water levels. Compared with litter removal, litter application increased the SOC by 25.12%, 9.58%, and 4.98% at the +25 cm, 0 cm, and -25 cm water levels, respectively. Hence, higher water levels facilitate the release of organic carbon from leaf litter into the soil via water leaching. In this way, they strengthen the soil carbon pool. At lower water levels, soil carbon is lost as the slower litter decomposition rate and active microbial (actinomycete) respiration. Our results revealed that the water level in natural wetlands influences litter decomposition mainly by leaching and microbial activity, by extension, affects wetland surface carbon pool.

**Key words:** *Carex brevicuspis*; decomposition; leaf litter; soil surface organic carbon pool; water level

**1 Introduction**

Wetlands are important terrestrial carbon pools. They contain $1.5 \times 10^3$ Pg carbon (~35% of the global carbon supply) and 25–63% of the soil carbon distributed in the 0–30 cm topsoil layer (Whiting and Chanton, 2001; Means et al., 2016; Cao et al., 2017). The surface soil organic carbon (SOC) pool (S-SOCP) and its turnover are sensitive to climate, topography, and hydrological condition (Wang et al., 2016; Zhang et al., 2017;Pinto et al., 2018).

Leaf litter decomposition is a major biotic carbon input route from vegetation to S-SOCP in wetland ecosystems (Whiting and Chanton, 2001; Moriyama et al., 2013). However, the reported impacts of litter decomposition on the soil carbon pool are highly variable (Busse et al., 2009; Crow et al., 2009). Litter input destabilised carbon storage by stimulating soil mineralisation and increasing soil labile carbon fractions (microbial biomass carbon [MBC] and soil dissolved organic carbon [DOC]) and enzyme activity in the freshwater marshland of Northeast China (Song et al., 2014). It also promoted soil carbon loss via $CO_2$ emissions and microbial activity in alpine and coastal wetlands (Gao et al., 2016; Liu et al., 2017). In contrast, a study has recently found that litter decomposition increased soil active organic carbon and stabilised the soil carbon pool in the Jiaozhou Bay wetland (Sun et al., 2019).



Litter decomposition is a physicochemical processes that reduces litter to its elemental chemical
constituents (Aerts, 1997). Litter decomposition rates are determined mainly by environmental factors
(climatic and soil conditions), litter quality (litter composition such as C, N, and lignin content) and
decomposer organisms (microorganisms and invertebrates) (Aerts, 1997). A previous study showed that
regional and global environmental conditions explain > 51% of the variation in litter decomposition rate
(Zhang et al., 2019). In wetland ecosystems, the water level ecosystem processes as it determines soil
aerobic and anaerobic conditions which, in turn, affect microbial decomposition of litter and SOC
decomposition (Liu et al., 2017; Yan et al., 2018). An earlier study reported that high soil moisture
content and long flooding periods facilitate litter decomposition by promoting leaching, fragmentation,
and microbial activity (Zhang et al., 2019). The water level may contribute to soil physicochemical
conditions which, in turn, regulate litter decomposition (Xie et al., 2016). Aboveground litter
decomposition is the main source of SOC (Upton et al., 2018). However, the contribution of litter
decomposition to the SOC pool has seldom been quantified.
Dongting Lake (28°30'–30°20' N, 111°40'–113°10' E) is the second largest freshwater lake in China. It
is connected to the Yangtze River via tributaries. Dongting Lake wetlands are characterised by large
seasonal fluctuations in water level ($\leq$ 15 m) and are completely flooded during June–October and
exposed during November–May (Chen et al., 2016). Peng et al. reported that the organic carbon density
in Dongting Lake wetland soil at 1 m depth was $127.3 \pm 36.1$ t hm$^{-2}$ and the carbon density in the 0–30 cm
topsoil was $46.5 \pm 19.7$ t hm$^{-2}$ (Peng et al., 2005). *Carex brevicuspis* is a dominant vegetation in the
Dongting Lake wetland and has large carbon reserves ($\sim 6.5 \times 10^6$ t y$^{-1}$) (Kang et al., 2009). However, due
to the dam construction in the upstream of Dongting Lake, the water regime varies remarkably (early
water withdrawn and declining of groundwater in non-flood season) in recent years, leading a significant
carbon loss in this floodplain wetland (Hu et al., 2018; Deng et al., 2018).
Here, we investigated *C. brevicuspis* litter decomposition and its contribution to the SOC pool at three
water levels (-25 cm, 0 cm, and +25 cm relative to the soil surface) to find the factors controlling *C.*
*brevicuspis* leaf litter decomposition and quantify the contribution of litter decomposition to the SOC
pool. We tested the following hypotheses. First, the litter decomposition rate at the +25 cm water level is
faster than that at the 0 and -25 cm water levels because of leaching, fragmentation, and microbial
activity. Second, the rates of litter carbon, nitrogen and phosphorus release are the highest at the +25 cm
water level, and the intrinsic litter decomposition controls differs among the three water levels because of





the differences among them regarding water infiltration. Third, the S-SOCP is relatively higher at the
+25-cm water level because of substantial carbon accumulation resulting from significant litter
decomposition. In contrast, it is comparatively lower at the -25 cm water level owing to carbon loss
because of slow litter decomposition.
**2 Materials and methods**
**2.1 Soil core collection and leaf litter preparation**
Soil cores (40 cm diameter × 50 cm length) and leaf litter were collected in May 2017 from an
undisturbed *Carex brevicuspis* community at the sampling site (29°27'2.02" N, 112°47'32.28" E) of the
Dongting Lake station for wetland ecosystem research. The litter was cleaned with distilled water,
oven-dried at 60 °C to a constant weight, and cut into pieces 5–10 cm long. Preweighed litter samples (5
g; $10.73 \pm 0.28$ g kg$^{-1}$ N, $0.89 \pm 0.04$ g kg$^{-1}$ P, $40.23 \pm 2.6\%$ organic C, and $17.83\pm0.25\%$ lignin) were
placed into 10 cm × 15 cm 1mm mesh nylon bags. This mesh size excluded macroinvertebrates but
permitted microbial colonisation and litter fragment leaching (Xie et al., 2016).
**2.2 Experimental design**
There were three water level treatments (-25 cm, 0 cm, and +25 cm relative to the soil surface) nested by
two litter treatments (input vs. removal) and three replicates. The experiment was conducted in nine
cement ponds (2 m × 2 m × 1 m) at the Dongting Lake station for wetland ecosystem research which is
part of the China Ecosystem Research Network. For the -25 cm treatment, the water level was 25 cm
below the soil surface. For the 0 cm treatment, the soil was fully wetted with belowground water but
without surface pooling. For the +25 cm treatment, the water level was 25 cm above the soil surface.
Water levels were adjusted weekly using belowground water (TOC: 3.44 mg L$^{-1}$; TN: 0.001 mg L$^{-1}$; TP:
0.018 mg L$^{-1}$). Three soil core sets were placed in each pond. One was designated the litter removal
control (S), the second was distributed on the soil surface in 15 litter bags to observe the effects of leaf
litter input on soil carbon pool (L), and the third was distributed on the soil surface in 15 litter bags to
monitor the litter decomposition rate and process (D) (Fig. 1). The experiment started on 20 August 2017
and lasted 280 d. By that time, no further significant change in litter dry weight was observed. Before
incubation, three litter and three soil samples were collected to determine their initial quality. Litter bags





were randomly collected from treatment D after 20 d, 40 d, 60 d, 80 d, 100 d, 130 d, 160 d, 190 d, 220 d,
250 d, and 280 d. After collection, the litter samples were separated, cleaned with distilled water, and
oven-dried at 60 °C to a constant weight (±0.01 g). All samples were pulverised and passed through a
0.5-mm mesh screen for litter quality analysis. At the end of incubation, the surface soil (0–5 cm, ~600 g
FW) was collected to eliminate the influences of root decomposition on the soil organic pool. The soil
samples were placed in aseptic sealed plastic bags and transported to the laboratory. The samples were
sieved (< 2 mm), thoroughly mixed, and divided into three subsamples. The first subsample (~150 g) was
stored at -20 °C and freeze-dried for phospholipid fatty acid (PLFA) analysis. The second one (~150 g)
was stored at 4 °C for MBC and DOC measurements. The third subsample (~300 g) was air-dried for
physicochemical analysis.

**2.3 Litter quality analyses**

Litter organic carbon content was analysed by the $H_2SO_4$-$K_2Cr_2O_7$ heat method. Litter nitrogen was
extracted by Kjeldahl digestion and quantified with a flow injection analyser (AA3; Seal Analysisten
GmbH, Langenselbold, Germany) (Xie et al., 2017). Litter phosphorus content was quantified by the
molybdenum-antimony anti-spectrophotometric method. The lignin content was measured by hydrolysis
(72% $H_2SO_4$) (Graça et al., 2005; Xie et al., 2017).

**2.4 Soil quality analyses**

**2.4.1 Soil chemical analyses**

SOC was determined by wet oxidation with $KCr_2O_7$ + $H_2SO_4$ and titration with $FeSO_4$ (Xie et al., 2017).
Soil DOC was extracted with $K_2SO_4$ and measured with a TOC analyser (TOC-VWP; Shimadzu Corp.,
Kyoto, Japan). MBC was analysed by chloroform fumigation, $K_2SO_4$ extraction, and TOC analyser
(TOC-VWP, Shimadzu Corp., Kyoto, Japan) (Tong et al., 2017).

**2.4.2 Soil microbial composition**

The total and specific microbial group biomass values and the microbial community structure were
estimated by phospholipid fatty acid (PLFA) analysis. The PFLAs were extracted from 8 g of
freeze-dried soil and analysed as previously described (Zhao et al., 2015). The concentrations of each
PLFA was calculated relative to that of the methyl nonadecanoate (19:0) internal standard. The PLFAs





for the following groups were determined: (a) bacterial biomass, sum of i15:0, a15:0, 15:0, i16:0, 16:1u7,
i17:0, a17:0, 17:0, cy17:0, and cy19:0; actinomycete biomass, sum of 10 Me 16:0, 10 Me17:0, and 10 Me
18:0; and fungal biomass, 18:2 ω6 and 18:1ω9. The total microbial biomass was represented by the sum
of the bacterial, fungal, and actinomycete biomass values. The ratios of fungal to bacterial lipids (F/B)
were used to evaluate the microbial community structure ((Bossio and Scow, 1998; Wilkinson et al.,
2002; Zhao et al., 2015).

### 2.5 Data processing

#### 2.5.1 Litter decomposition rate

The percentage of litter dry weight loss was calculated as follows (Zhang et al., 2019):
$L_t = \frac{M_0 - M_t}{M_0} \times 100\%$ (1)
where $L_t$ is the percentage litter dry weight loss at time $t$ (%), $M_t$ is the litter dry matter weight at the time
$t$ (g), and $M_0$ is the initial dry matter weight (g).
The instantaneous litter dry mass decay rate ($k$) was calculated using the Olson negative exponential
attenuation model (Olson, 1963):
$M_t = M_0 e^{-kt}$        (2)
where $M_t$ is the litter dry matter weight at time $t$ (g), $M_0$ is the initial dry matter weight (g), $k$ is the
instantaneous litter decay rate at time $t$, and $e$ is the natural base number. $M_t$ increased with litter
decomposition rate.

#### 2.5.2 Relative release index

The relative release indices (RRIs) of C, N, and P from the plant litter were calculated as follows (Zhang
et al., 2019):
$RRI_t = \frac{M_0 \times C_0 - M_t \times C_t}{M_0 \times C_0} \times 100\%$    (3)
where $C_t$ is the concentration of an element in the litter at time $t$, $C_0$ is the initial concentration of an
element in the litter, and $M_t$ is the litter dry matter weight at time $t$ (g). CRRI, NRRI, PRRI, and LRRI
represent the carbon, nitrogen, phosphorus, and lignin RRIs, respectively. A positive RRI indicates a net
release of the element during litter decomposition whilst a negative RRI indicates a net accumulation of
the element during litter decomposition.





### 2.5.3 Contribution of litter-C input to the SOC pool

The contribution of litter-C input to the SOC pool was calculated as follows (Lv and Wang, 2017):

$$LC = \frac{SOC_L - SOC_S}{SOC_i} \times 100\% \qquad (4)$$

where $LC$ is the contribution of the litter-C input to SOC pool, $SOC_L$ is the SOC concentration for the

litter input treatment, $SOC_S$ is the SOC concentration for the treatment without litter input, and $SOC_i$ is

the initial SOC content before the experimental treatments.

### 2.6 Statistical analyses

The percentage litter dry weight losses and the instantaneous decomposition rates were compared among

the three water levels by repeated ANOVA. Water level was the main factor and time was the repeated

factor. The surface soil chemical components and the microbial community structure were compared by

two-way ANOVA. Treatment (with or without litter input) and water level were the main factors. The

percentage differences in litter dry weight loss, the instantaneous decomposition rates, the soil chemical

components, and the microbial community structure were evaluated by LSD at the 0.05 significance

level. The data were expressed as means ± standard error. All statistical analyses were performed in

SPSS 21 (IBM Corp., Armonk, NY, USA).

### 3 Results

### 3.1 litter decomposition process

The percentage litter dry weight loss was the highest for the +25 cm water level treatment through the

entire litter decomposition period followed by the 0 cm water level treatment. The percentage litter dry

weight loss was the lowest for the -25 cm water level treatment ($P < 0.01$; Fig. 2A). After 280 d

decomposition, the percentage litter dry weight loss values under the +25 cm, 0 cm, and -25 cm water

level treatments were 61.8%, 49.8% and 32.4%, respectively.

The instantaneous decomposition rate at each measurement time point was calculated using the Olson

negative exponential decay model. The instantaneous litter dry weight decomposition rate rapidly

increased and then slowly decreased and stabilised for all three water levels. The maximum

decomposition rates for the -25 cm, 0 cm, and +25 cm water levels were 0.0069631 $d^{-1}$, 0.008990 $d^{-1}$, and

0.012954 $d^{-1}$, respectively (Fig. 2B).





**3.2 Intrinsic litter decomposition rate-limiting factor**

During the entire decomposition process, CRRI, NRRI, PRRI, and LRRI significantly increased with water level. Litter carbon and lignin were always released at all three water levels whilst at -25 cm, nitrogen and phosphorus enrichment appeared in the middle stage (Fig. 3A–3D). At the start of the experiment, neither the C/N nor the lignin/N ratio significantly differed at the three water levels. At the middle stage, however, both the C/N and lignin/N ratios were significantly lower at the -25-cm water level than they were at the 0 cm and -25 cm water levels (Fig. 3E–3F).

The multiple regression model of the instantaneous litter decomposition rate and the litter properties showed that at the -25 cm and 0 cm water levels, the main decomposition rate-limiting factor was the lignin concentration whilst at the +25 cm water level, the main litter decomposition rate-limiting factor was the lignin/N ratio (Table 1).

**3.3 Soil surface microbial community structure**

Under both litter input and litter removal conditions, the bacterial, fungal, and microbial biomass levels were the highest under the 0 cm water level treatment; however, these parameters showed nonsignificant differences between 25 cm above and below water level treatments ($P > 0.05$; Fig. 4A, 4B, and 4F). The actinomycete biomass was the highest under the -25 cm water level treatment, followed by that under the 0 cm water level treatment. It was the lowest under the +25 cm water level treatment (Fig. 4C). Litter input significantly stimulated fungal and microbial biomass at all three water levels but only significantly stimulated bacterial and actinomycete biomass at the -25 cm water level ($P < 0.05$; Fig. 4A–4C and 4E). Under litter input conditions, the fungal/bacteria ratio was the highest at the 0 cm water level, followed by the +25 cm water level. It was the lowest under the -25 cm water level treatment. Under litter removal conditions, however, the fungal/bacteria ratio was significantly higher under the -25 cm water level treatment than it was under the 0 cm and +25 cm water level treatments ($P < 0.05$; Fig. 4D).

**3.4 Contribution of leaf decomposition to the soil surface carbon pool**

The SOC, MBC, and DOC concentrations were significantly affected by the water level. SOC and MBC were the highest at the 0 cm water level and the lowest at the -25 cm water level ($P < 0.01$; Fig. 5A and 5B). DOC was the highest at the -25 cm water level and the lowest at the +25 cm water level ($P < 0.01$; Fig. 5C).





Compared with the litter removal group, the SOC concentrations were significantly higher for the litter
input group at the +25 cm and 0 cm water levels. Relative to the litter removal group, the DOC
concentrations were significantly higher for the litter input group at the 0 cm- and -25 cm water levels ($P$
$< 0.001$; Fig. 5A and 5C). The contribution of the litter-C input to the S-SOCP was the highest for the +25
cm water level treatment (13.75%), intermediate for the 0 cm water level treatment (4.73%), and the
lowest for the -25 cm water level treatment (2.51%) ($P < 0.001$; Fig. 5D).
**4 Discussion**
**4.1 Environmental control of litter decomposition**
Water level significant influenced *C. brevicuspis* leaf litters decomposition ($P < 0.001$). The *K*-values
were the highest for the +25 cm water level treatment, intermediate for the 0 cm water level treatment,
and the lowest for the -25 cm water level treatment (Fig. 2B). Hence, the percentage litter dry weight loss
and the decomposition rate increased with water level. These results supported our first hypothesis. The
wetland water level strongly affect litter leaching and microbial decomposition (Peltoniemi et al., 2012).
Compared with the terrestrial environment, water promotes litter leaching and microbial metabolism,
thereby accelerating litter decomposition. Moreover, water infiltration into litter also increases relative
leaching loss (Molles et al., 1995). Here, the high litter decomposition rate measured for the +25 cm
water level treatment may be explained primarily by litter leaching. This finding was consistent with
those reported for *Carex cinerascens* litter decomposition in Poyang Lake (Zhang et al., 2019) and
*Calamagrostis angustifolia* litter decomposition on the Sanjiang Plain (Sun et al., 2012).
The high soil total microbial, bacterial and fungal biomass levels at the 0 cm water level could account
for the rapid litter decomposition observed there. Certain microorganisms are vital to the decomposition
process (Yarwood, 2018). Fungi are primary litter decomposers as they fragment dead plant tissues by
breaking down lignin and cellulose. Bacteria are secondary decomposers that utilise the simpler
compounds generated by fungal activity (de Boer et al., 2005; Bani et al., 2019). Microbial decomposers
generally flourish in humid environments. At the 0-cm water level, microbial activity explains most of
the litter decomposition. At the -25-cm water level, the soil surface is dry, there are comparatively few
microbial decomposers, and decomposition is very slow.





**4.2 Intrinsic factors controlling litter decomposition**

Decomposition rapidly increased to reach a maximum by 20 d. Thereafter, it slowly decreased and then stabilised (Fig. 2B). Water-soluble components and non-lignin carbohydrates are preferentially and quickly decomposed at the onset of decomposition (Davis et al., 2003). Here, a multiple regression model of the instantaneous litter decomposition rate and litter properties showed that the internal limiting factors affecting the rate of *C. brevicuspis* leaf litter decomposition varied with water level. The lignin concentration determined the litter decomposition rate for the -25 cm and 0 cm water level treatments whilst the lignin/N ratio regulated the litter decomposition rate for the +25 cm water level treatment. This discovery upheld our second hypothesis and was consistent with the findings of Zhang et al. who reported that wetland ecosystems decomposed *Carex cinerascens* lignin much earlier faster than did terrestrial ecosystems (Zhang et al., 2019). Here, we found that the lignin content was the major internal limiting factor of the *C. brevicuspis* leaf litter decomposition rate. At the +25 cm water level, N is rapidly lost and the L/N ratio significantly increases. Thus, L/N is the main internal limiting factor at the +25 cm water level. A few studies showed that the lignin content is a key factor limiting terrestrial plant and hygrophyte litter decomposition (Yue et al., 2016; Zhang et al., 2018). Therefore, the amount of carbon that the litter can return to the ecosystem is closely associated with the plant lignin content. The lignin content of *C. brevicuspis* leaf litters is ~10% less than that of other wetland plants such as *Miscanthus sacchariflorus* (~30%) (Xie et al., 2016), *Spartina alterniflor* (~40%) (Yan et al., 2019), and terrestrial plants such as willow (~25%), larch (~38%), and cypress (~28%) (Yue et al., 2016). Furthermore, *C. brevicuspis* covers a large area (~23,950 $hm^2$) and generates abundant litter (~36,547 t) (Kang et al., 2009). Thus, *C. brevicuspis* litter may potentially return large amounts of carbon to the soil.

**4.3 Contribution of leaf decomposition to the soil surface carbon pool**

Litter decomposition is the main pathway by which nutrients are transferred from the plants to the soil. Litter affects SOC whose stabilisation affects other soil properties such as sorption, nutrient availability, pH, and water holding capacity (Brady and Weil, 2008). The results of this study showed that litter addition increases SOC in a manner that varies with water level. The contribution of litter-C input to the S-SOCP was the highest under the +25 cm water level treatment (13.75%), intermediate under the 0 cm water level treatment (4.73%), and the lowest under the -25 -cm water level treatment (2.51%). For this reason, flooding conditions are conducive to litter carbon input into the soil. These findings corroborated





our third hypothesis. In addition, litter input had a similar effect on soil DOC at the 0 cm and -25 cm
water levels. Therefore, litter decomposition contributes mainly soluble carbon to the soil (Zhou et al.,
2015). However, this DOC is also readily lost and decomposed (Sokol and Bradford,
2019;Gomez-Casanovas et al., 2020). This fact accounts for the significantly lower relative DOC under
the +25 cm water level treatment here. Wetlands have comparatively larger but also more unstable
S-SOCPs than terrestrial environments. In wetlands, water level fluctuations could readily cause carbon
loss (Gao et al., 2016; Chen et al., 2018). Nevertheless, we considered mainly aboveground litters in this
experiment. Hence, the influences of underground litter (root) decomposition on the SOC pool should be
investigated in future research (Sokol and Bradford, 2019; Lyu et al., 2019).

**5 Conclusion**

The water level in natural wetlands influences litter decomposition mainly by leaching and microbial
activity, by extension, affects wetland surface carbon pool. Higher water levels facilitate the release of
organic carbon from leaf litter into the soil via water leaching, and thus strengthened soil carbon pool. At
lower water levels, wetland soil carbon is lost as the litter decomposition is lower, but active microbial
(actinomycete) respiration rates is comparatively higher. The groundwater decline which was caused by
the climate change and human disturbance in Dongting Lake floodplain would slowdown the return
rate of organic carbon from leaf litter to soil, and facilitate the S-SOCP loss. Therefore, in wetland
management and restoration practices, the construction of microhabitat with prolonged flooding period
and relatively higher water level are essential ways to improve the carbon sequestration potential.

**Data availability**

The data used in this paper are stored in the open-access online database Figshare and can be accessed
using the following link: https://doi.org/10.6084/m9.figshare.12758387.v1 (Zhu et al. 2020).

**Conflict of interest**

The authors declare that they have no conflict of interest.



**Author contributions**

Lianlian Zhu designed experiments, collected samples, acquired, analyzed, interpreted data, and wrote the manuscript. Zhengmiao Deng designed experiments, interpreted data and revised the manuscript. Yonghong Xie designed experiments and revised the manuscript. Xu Li, Feng Li, Xinsheng Chen and Yeai Zou collected samples and revised the manuscript. Chengyi Zhang and Wei Wang interpreted data and revised the manuscript.

**Acknowledgement**

We are immensely thankful to teachers in Dongting Lake Station and Key Laboratory of Agro-ecological Processes in Subtropical Region for their help in soil sample collection and chemical analysis. We would like to thank Editage [www.editage.cn] for English language editing. The data used in this paper are available in the supplement material.

**Financial support**

This study was financial supported by the Hunan innovative province construction projection (Hunan Key Research and Development Project, 2019NK2011), the National Natural Science Foundation of China (41401290), Changsha Natural Science Funds for Distinguished Young Scholar (2020), the Natural Science Foundation of Hunan province(2018JJ3580).

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





**Table 1: Multiple regression model of instantaneous litter decomposition rate and litter properties**

| Water level (cm) | Multiple regression model | $F$ | $R^2$ | $P$ |
|---|---|---|---|---|
| -25 | $R = -\mathbf{0.861L} - 0.404C + 0.239CN + 2.586$ | 27.264 | 0.887 | < 0.001 |
| 0 | $R = -\mathbf{1.131L} - 0.390P - 0.218LN + 2.124$ | 48.330 | 0.934 | < 0.001 |
| +25 | $R = -\mathbf{0.739LN} - 0.636N + 4.162$ | 19.465 | 0.787 | < 0.001 |

**where *R* is the litter instantaneous decomposition rate, L is the lignin concentration, C is the carbon**
**concentration, N is the nitrogen concentration, P is the phosphorus concentration, CN is the**
**carbon-to-nitrogen ratio (C/N, g g⁻¹), and LN is the lignin-to-nitrogen ratio (lignin/N, g g⁻¹).**



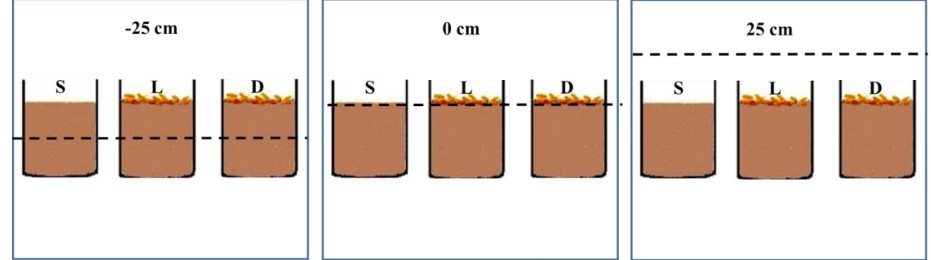


**Figure 1: Schematic diagram of the experimental setup. The dotted line represents the water level.**

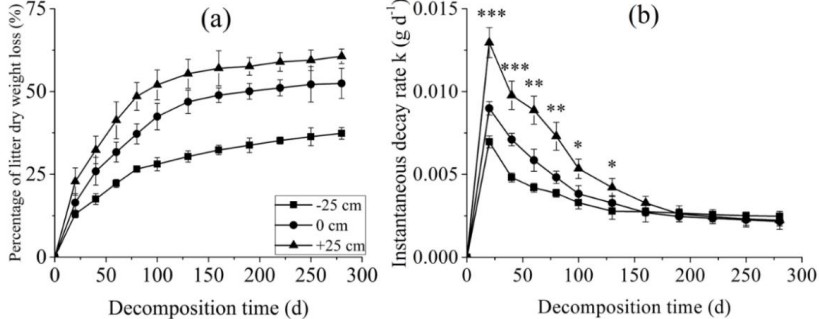


**Figure 2: Percentage litter dry weight loss and decomposition rate during *C. brevicuspis* decomposition at**
**three water levels (-25 cm, 0 cm, and +25 cm). *, **, and *** represent significant differences of the litter**
**instantaneous decay rate among the three water levels at the 0.05, 0.01, and 0.001 significance levels,**
**respectively.**

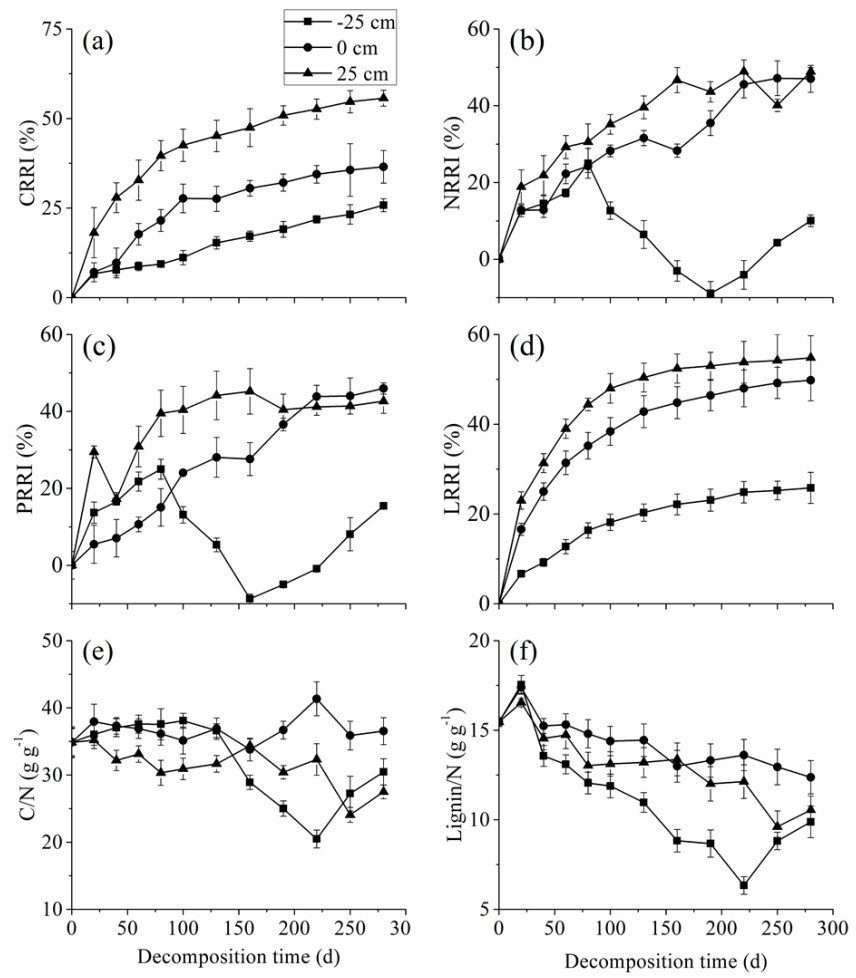

**Figure 3: Percentage (mean ± SE) of carbon relative release index (CRRI), nitrogen relative release index (NRRI), phosphorus relative release index (PRRI), lignin relative release index (LRRI), C/N ratio, and lignin/N ratio at three water levels (-25 cm, 0 cm, and +25 cm).**

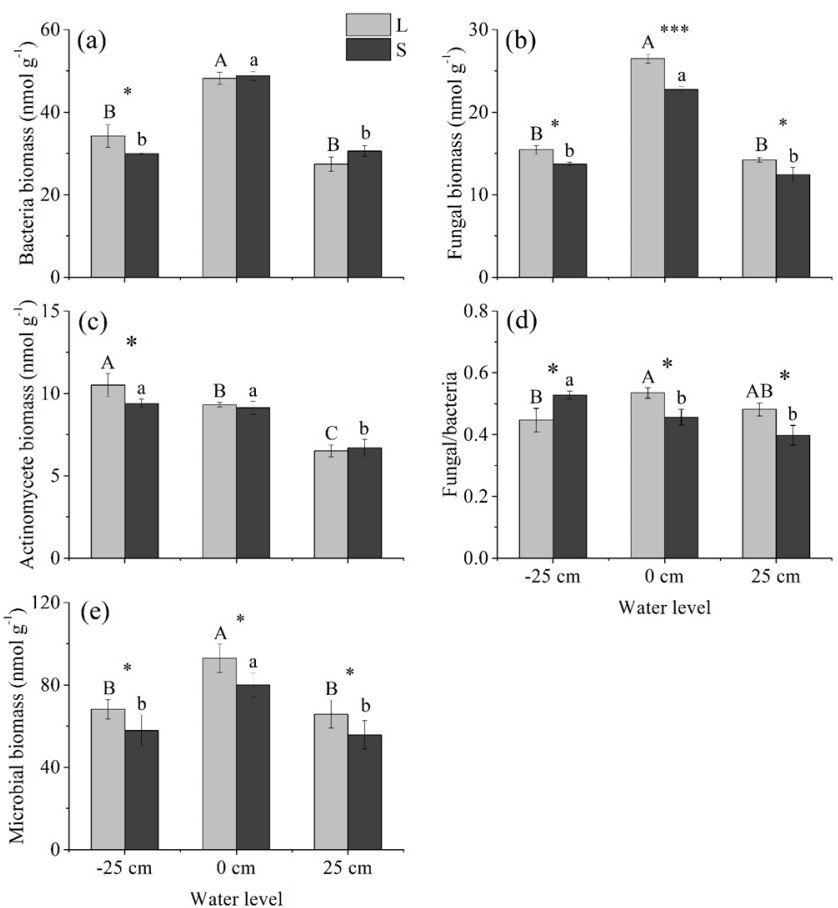

**Figure 4: Microbial community structure under litter input and litter removal at three water levels. Different uppercase letters among vertical bars indicate significant differences among the three water levels in the litter input (L) group. Different lowercase letters indicate significant differences among the three water levels in the litter removal (S) group. The significance level is α = 0.05. *, **, and *** represent significant differences between the litter input (L) and litter removal (S) groups at the three water levels at the 0.05, 0.01, and 0.001 significance levels, respectively.**


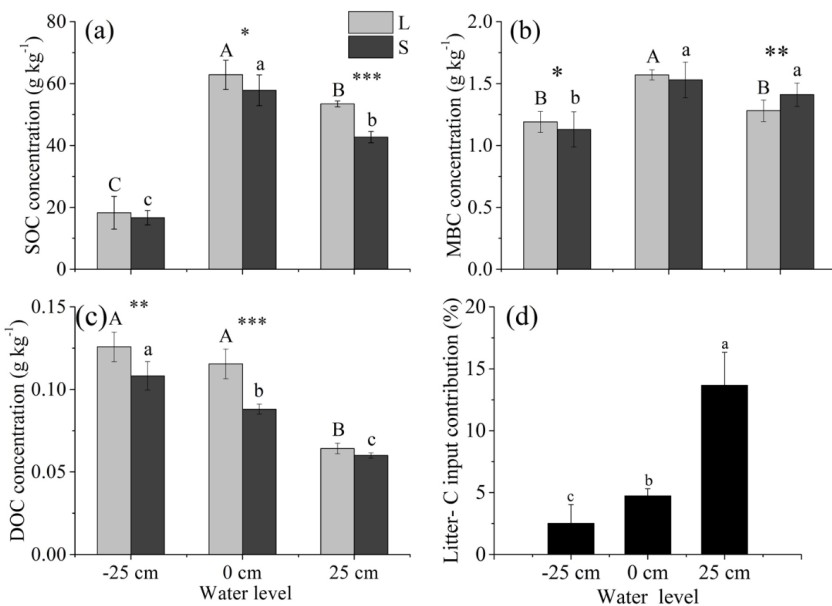

481

**Figure 5: Concentrations of SOC (A), MBC (B), DOC (C) between the litter input (L) and litter removal (S) groups and the litter-C input contribution (D) under three water levels at the end of the experiment. Different uppercase letters among vertical bars indicate significant differences among the three water levels in the litter input (L) group. Different lowercase letters indicate significant differences among the three water levels in the litter removal (S) group. The significance level is α = 0.05. \*, \*\*, and \*\*\* represent significant differences between the litter input (L) and litter removal (S) groups at the three water levels at the 0.05, 0.01, and 0.001 significance levels, respectively.**