# Peer review of "decomposition and its contribution to surface soil organic"

_Biogeosciences, 2020_

## Referee Comment (RC1) · Anonymous Referee #1 · 12 Aug 2020

General comments: Zhu et al. not only identified the major factor controlling leaf litter decomposition as water level, but also revealed its working approach in natural freshwater wetlands. The systematic and scientifically sound design delivered new insights into wetland leaf litter decomposition processes and consequences. I recommend to be accepted after revision.

Specific comments: Abstract: L25-27: The key rate values should be added. L33: Change "strengthen" to "increase". L35: Change "influences" to "influenced". L36: Change "affects" to "and affected".

Introduction: L40: Change "25" to "25%". L66-69: Move to M & M. L71: Species is not

a vegetation. L82: Unclear. "decomposition controls differs"?

Materials and methods: L100: Move "which is ..." to L91. L101: What's the source of the belowground water? L105: How to arrange the 15 litterbags (10 cm × 15 cm) within each soil cores (40 cm diameter)? L170: Multiple regression method should be added.

Results L198-201, Table 1: Why not choose the same variables in every regression model? Please explain or give the methodology basis. Figure 1: The full words of S, L and D should be added in the caption.

Discussion L227: K-value should be kept consistent with k occurred in M & M. L230: Please specify which results. L232-233, 244-245: Not always the truth. Water will inhibit most decomposition as well for lack of oxygen. L251-259: It's more interesting to discuss why the same litter subject to various water levels were mainly controlled by different factor? L279-280? Any references?

Conclusion L285-286: Repeated from Abstract. L291-293: Beyond the support of this study.

---

## Referee Comment (RC2) · Anonymous Referee #2 · 1 Sep 2020

Line 20 and 46: I recommend including (Cao et al. 2020), and references therein, that address aboveground litter decomposition on SOC pools.

Line 30: The SOC increase due to litter application (Figure 5d) appears to be calculated from Figure 5a, but I could not reconcile the value for -25 cm. While Figure 5 does seem to support that litter increases SOC, I have two concerns about this presentation. First, the differences in Figure 5a for the 0 cm water levels is labelled as significant, but the error bars clearly overlap. Please clarify. Second, and a potential fundamental flaw in the data presentation, is that the baseline SOC is not provided and there is no way to know if SOC changed throughout the course of the experiment. It appears based on

[Figure]

Equation 4 you did measure baseline SOC for each core ($\sim$18 g kg-1)? Please clarify. In either case, a significant observation, which is not discussed, is that the no-litter treatments resulted in very large SOC increases and adding litter resulted in a small additional increase.

Line 40: References for this statement are inappropriate, or incorrectly cited. Means et al. 2016 does not discuss global carbon pools. Whiting and Chanton (2001) is an accurate source for the value you used for wetland carbon stocks, but they cite Schlesinger 1991 as their source, and there are more up-to-date carbon stock estimates, such as (Köchy, Hiederer, and Freibauer 2015). Cao et al., 2017, is a secondary reference, like Whiting and Chanton (2001), and neither reflect the range of wetland soil carbon (25 – 63%) you provide. The value in Whiting and Chanton (2001) is 3 – 68% (secondary references) and the value in Cao et al. (2017) is 12 – 15% (also a secondary reference). Use of the most up-to-date sources and an accurate reflection of those sources adds value to the manuscript. I recommend adding a citation such as (Kayranli et al. 2010), which could also be useful in your discussion considering what happens to the SOC after it is leached from the litter into the soil.

Line 52: The Sun et al., 2019 study does not support the statement that litter decomposition stabilized the soil organic carbon pool. Litter decomposition made DOC more mobile and labile, which the authors suggested could lead to SOC stability after processing by soil microbes. Lines 54-56: Aerts (1997) addresses litter decomposition in non-wetland sites where shredder invertebrates (detritovores) are important, but their role in wetland settings is more uncertain (Inkley, Wissinger, and Baros 2008). Shredding would be an important physio-chemical control on DOC leaching.

Line 62: Zhang 2019 supports the statement that water levels affected microbial activity, but leaching and fragmentation were only discussed, not measured.

Line 64: This is a mischaracterization of the Upton, 2018 study. Perhaps a better reference is (Hoyos-Santillan et al. 2015). However, clarification is needed because

Hoyos-Santillan states that roots (not litter) are the main source of SOC in peatlands, but litter strongly influences root decomposition rates, particularly near the surface.

Line 151. The Olson (1963) simple decay model assumes constant k, which you demonstrated is not a constant. Although use of this decay model is common in the literature, it is an oversimplification. This does not adversely affect your comparative analysis, but the paper would be strengthened with a more sophisticated analysis, such as a double exponential decay model (Berg 2014 or Wider and Lang 1982).

Line 266: Doesn't your argument imply C. brevicuspis, due to it's lower lignin content, would return less carbon to the soil compared to the other plants you cited? The manuscript may be strengthened, and have a wider inference, if you listed and compared decomposition rates and lignin contents of wetland plants including C. brevicuspis.

Line 284: Your conclusion contains a significant amount of new and largely unsupported discussion material. Conclusions should stick to what you were able to show in your experiments.

Table 1: Is the lignin content (L) in the regression model the initial lignin content?

Figure 3d (LRRI%) is nearly identical to Figure 2a (litter dry weight loss, %), which seems counter-intuitive unless lignin were the sole material being mineralized during the decomposition process. Did you measure lignin content at each time point?

technical corrections

Line 23: Carex brevicuspis may be ubiquitous to wetlands in China; however, is this true globally?

Line 25: Is "mass loss" = "carbon release"? If so, one of these phrases is redundant.

Line 82: The way you stated your hypotheses imply you have tested causal factors, which you did not. Specifically, leaching, fragmentation and infiltration.

Line 102: Did the litter bags float? Did you need to pin them in contact with the soil surface?

Line 105 - 107: Clarify how many soil cores were used in each pond for each purpose and how they were prepared (e.g. were soils blended prior to starting the experiment). The text is confusing.

Line 183: You use capital letters (Fig 2A) in text references, but lower-case letters in the Figures.

Line 187/188 and Line 247: I would not interpret your data that decomposition rates "rapidly increased" – the decomposition rate at time t=0 is undefined.

Figure 4: Are these figures reporting mass? The units are nmol g-1.

Berg, Björn. 2014. "Decomposition Patterns for Foliar Litter – A Theory for Influencing Factors." Soil Biology and Biochemistry 78 (November): 222–32. https://doi.org/10.1016/j.soilbio.2014.08.005. Cao, Jianbo, Xinxing He, Yuanqi Chen, Yuping Chen, Yanju Zhang, Shiqin Yu, Lixia Zhou, Zhanfeng Liu, Chenlu Zhang, and Shenglei Fu. 2020. "Leaf Litter Contributes More to Soil Organic Carbon than Fine Roots in Two 10-Year-Old Subtropical Plantations." Science of The Total Environment 704 (February): 135341. https://doi.org/10.1016/j.scitotenv.2019.135341. Hoyos-Santillan, Jorge, Barry H. Lomax, David Large, Benjamin L. Turner, Arnoud Boom, Omar R. Lopez, and Sofie Sjögersten. 2015. "Getting to the Root of the Problem: Litter Decomposition and Peat Formation in Lowland Neotropical Peatlands." Biogeochemistry 126 (1–2): 115–29. https://doi.org/10.1007/s10533-015-0147-7. Inkley, Martyn D., Scott A. Wissinger, and Brandi L. Baros. 2008. "Effects of Drying Regime on Microbial Colonization and Shredder Preference in Seasonal Woodland Wetlands." Freshwater Biology 53 (3): 435–45. https://doi.org/10.1111/j.1365-2427.2007.01908.x. Kayranli, Birol, Miklas Scholz, Atif Mustafa, and Åsa Hedmark. 2010. "Carbon Storage and Fluxes within Freshwater Wetlands: A Critical Review." Wetlands 30 (1): 111–24. https://doi.org/10.1007/s13157-009-0003-4. Köchy, M., R.

[Figure]

Hiederer, and A. Freibauer. 2015. "Global Distribution of Soil Organic Carbon – Part 1: Masses and Frequency Distributions of SOC Stocks for the Tropics, Permafrost Regions, Wetlands, and the World." SOIL 1 (1): 351–65. https://doi.org/10.5194/soil-1-351-2015. Wider, R. Kelman, and Gerald E. Lang. 1982. "A Critique of the Analytical Methods Used in Examining Decomposition Data Obtained From Litter Bags." Ecology 63 (6): 1636. https://doi.org/10.2307/1940104. Zhang, Quanjun, Guangshuai Zhang, Xiubo Yu, Yu Liu, Shaoxia Xia, Li Ya, Binhua Hu, and Songxian Wan. 2019. "Effect of Ground Water Level on the Release of Carbon, Nitrogen and Phosphorus during Decomposition of Carex. Cinerascens Kükenth in the Typical Seasonal Floodplain in Dry Season." Journal of Freshwater Ecology 34 (1): 305–22. https://doi.org/10.1080/02705060.2019.1584128.

---

## Author Comment (AC1) · 18 Sep 2020

Comment 1.1 General comments Zhu et al. not only identified the major factor controlling leaf litter decomposition as water level, but also revealed its working approach in natural freshwater wetlands. The systematic and scientifically sound design delivered new insights into wetland leaf litter decomposition processes and consequences. I recommend to be accepted after revision. Response 1.1 We appreciate the positive evaluations from the reviewer on our work and are grateful for the reviewer for recognizing the potential impact of our work. Comment 1.2 Specific comments Abstract: L25-27: The key rate values should be added. Response 1.2 Thank you very much

for your detailed suggestion. L25-27 has been changed to: The percentage litter dry weight loss and the instantaneous litter dry weight decomposition rate were the highest at +25 cm water level (61.8%, 0.01307d-1), followed by the 0 cm water level (49.8%, 0.00908 d-1), and the lowest at -25 cm water level(32.4%, 0.00527 d-1). See line 25-27 in the revised manuscript. Comment 1.3 L33: Change "strengthen" to "increase". Response 1.3 Changed as suggested, Thank you! Comment 1.4 L35: Change "influences" to "influenced". Response 1.4 Changed as suggested, Thank you! Comment 1.5 L36: Change "affects" to "and affected". Response 1.5 Changed as suggested, Thank you! Comment 1.6 Introduction: L40: Change "25" to "25%". Response 1.6 Changed as suggested, Thank you! Comment 1.7 L66-69: Move to M & M. Response 1.7 Thank you for the suggestion. We have moved "Dongting Lake (28°30'–30°20' N, 111°40'–113°10' E) is the second largest freshwater lake in China. It is connected to the Yangtze River via tributaries. Dongting Lake wetlands are characterized by large seasonal fluctuations in water level ($\leq$ 15 m) and are completely flooded during June–October and exposed during November–May (Chen et al., 2016)." to "Materials and methods" section as suggested. Please see L82-L85. Comment 1.8 L71: Species is not a vegetation Response 1.8 Thank you for the reminder. This sentence has been rephrased as "Carex brevicuspis is a dominant species in the Dongting Lake wetland". Please see L69. Comment 1.9 L82: Unclear. "decomposition controls differs"? Response 1.9 We are sorry for the ambiguity. It means that the intrinsic control factors are different at different water levels. This sentence has been rephrased as "the intrinsic factors that control litter decomposition rate at three water levels are different" to avoid confusing. Please see L78-79. Comment 1.10 L100: Move "which is . . ." to L91. Response 1.10 Thank you for the comment. This sentence has been moved to line 88-89 in the revised manuscript as suggested. Comment 1.11 L101: What's the source of the belowground water? Response 1.11 We are very sorry for our negligence of detailed description about the belowground water. The belowground water is extracted from the well in the experiment site by the water pump. We have added this information in M&M part. Please see L99-100. Comment

1.12 L105: How to arrange the 15 litterbags (10 cm × 15 cm) within each soil cores (40 cm diameter)? Response 1.12 We are very sorry for our negligence. Litter bags were laid flat on the surface of the soil. Each litter bag was not filled, and there are a little overlap between the litter bags where there is no litter. Please see L104-105. Comment 1.13 L170: Multiple regression method should be added. Response 1.13 Thank you for the suggestion. We have added the following sentence in the statistical analyses section. The intrinsic litter decomposition rate-limiting factors were analyzed by stepwise regression method in a multiple regression model. Please see L174-175. Comment 1.14 L198-201, Table 1: Why not choose the same variables in every regression model? Please explain or give the methodology basis. Response 1.14 We are very sorry for our negligence. Stepwise regression is used to calculate the regression model, the variables were the result of Stepwise regression model filtering, so the variables were different. The regression model methodology has been added in section 2.6, in L174-175. Comment 1.15 Figure 1: The full words of S, L and D should be added in the caption. Response 1.15 Thank you for the detailed suggestion. The following paragraph has been added in the Figure 1 caption. L represents litter which was distributed on the soil surface in 15 litter bags to observe the effects of leaf litter input on soil carbon pool; S represents soil which was designated the litter removal control; D represents decomposition which was distributed on the soil surface in 15 litter bags to monitor the litter decomposition rate and process. Please see L454-458. Comment 1.16 L227: K-value should be kept consistent with k occurred in M & M. Response 1.16 We are very sorry for our negligence. K-value in L227 in the original manuscript has been changed to the instantaneous litter dry mass decay rate (k) that was kept consistent with k occurred in M & M. Please see L229-230. Comment 1.17 L230: Please specify which results. Response 1.17 These results are that the percentage litter dry weight loss and the decomposition rate increased with water level supported our first hypothesis. "Hence, the percentage litter dry weight loss and the decomposition rate increased with water level. These results supported our first hypothesis." has been rephrased as following: "Hence, the percentage litter dry weight

loss and the decomposition rate increased with water level, which supported our first hypothesis." Please see L232-233. Comment 1.18 L232-233, 244-245: Not always the truth. Water will inhibit most decomposition as well for lack of oxygen. Response 1.18 We are sorry for the ambiguity. The purpose of quoting this sentence is to show that there are existing studies supporting the results in our study, and to provide a scientific and reasonable explanation for my research results. The sentences have been changed to: Related research showed that the wetland water level strongly affects litter leaching and microbial decomposition (Peltoniemi et al., 2012). Molles et al (1995) also found that compared with the terrestrial environment, in wetland, water promotes litter leaching and microbial metabolism, thereby accelerating litter decomposition. Moreover, water infiltration into litter also increases relative leaching loss (Molles et al., 1995). Please see L234-238. Comment 1.19 L251-259: It's more interesting to discuss why the same litter subject to various water levels were mainly controlled by different factor? Response 1.19 Thank you for the suggestion. This is mainly because that in different water levels, the rates of N lost were different. At the 0 cm and +25 cm water level, N is rapidly lost and the L/N ratio significantly increases. Thus, L/N is the main internal limiting factor at the 0 cm and +25 cm water level. Please see L261-263. Comment 1.20 L279-280? Any references? Response 1.20 References (Gao et al., 2016;Chen et al., 2018) have been added in the text. Please see L286-287. Comment 1.21 L285-286: Repeated from Abstract. Response 1.21 We are sorry for the mistake. The conclusion has been rephrased as "In this study, we quantified the contribution of leaf litter decomposition on soil surface organic carbon pools (S-SOCPs) under different water level conditions. Appropriate flooding (+25 cm water level treatment in our study) can significantly promote the decomposition of litters and contributed about 13.75% organic carbon to S-SOCPs. Under waterlogging condition (0 cm water level), litter decomposition, which mainly controlled by microbial activity, contributed 4.73% organic carbon to S-SOCP. However, under relatively drought condition (-25 cm water level treatment in our study), litter decomposition only contribute about 2.51% organic carbon to S-SOCP, which is largely ascribe to

the slower decomposition rate and soil carbon lost by metabolism of the microbes (i.e. actinomycete). We also found that lignin and/or lignin/N content were intrinsic factors controlling litter decomposition rate in Carex brevicuspis. In Dongting Lake floodplain, the groundwater decline which was caused by the climate change and human disturbance would slowdown the return rate of organic carbon from leaf litter to soil, and facilitate the S-SOCP loss." Please see L291-302. Comment 1.22 L291-293: Beyond the support of this study. Response 1.22 We accept this comment and this part had been deleted. Our conclusion has been rephrased as follows: Appropriate flooding can promote the decomposition of litters. The flooding can break the limit of lignin in the decomposition of litters. The flooding state is more conducive to the input of litter carbon into the soil, and the main input form may be DOC. The references in the responses were listed as follows: Chen, H. Y., Zou, J. Y., Cui, J., Nie, M., and Fang, C. M.: Wetland drying increases the temperature sensitivity of soil respiration, Soil Biology & Biochemistry, 120, 24-27, 10.1016/j.soilbio.2018.01.035, 2018. Gao, J. Q., Feng, J., Zhang, X. W., Yu, F. H., Xu, X. L., and Kuzyakov, Y.: Drying-rewetting cycles alter carbon and nitrogen mineralization in litter-amended alpine wetland soil, Catena, 145, 285-290, 10.1016/j.catena.2016.06.026, 2016.

Please also note the supplement to this comment:
https://bg.copernicus.org/preprints/bg-2020-266/bg-2020-266-AC1-supplement.pdf

———————————————————

[Figure]

**Fig. 1.** Figure 1: Schematic diagram of the experimental setup. The dotted line represents the water level. L represents litter which was distributed on the soil surface in 15 litter bags to observe the effect

[Figure]

[Figure]

**Fig. 2.** Figure 2: Percentage litter dry weight loss and decomposition rate during C. brevicuspis decomposition at three water levels (-25 cm, 0 cm, and +25 cm). *, **, and *** represent significant difference

[Figure]

**Fig. 3.** Figure 3: Percentage (mean ± SE) of carbon relative release index (CRRI), nitrogen relative release index (NRRI), phosphorus relative release index (PRRI), lignin relative release index (LRRI), C/N ra

[Figure]

**Fig. 4.** Figure 4: Microbial community structure under litter input and litter removal at three water levels. Different uppercase letters among vertical bars indicate significant differences among the three wa

[Figure]

**Fig. 5.** Figure 5: Concentrations of SOC (A), MBC (B), DOC (C) between the litter input (L) and litter removal (S) groups and the litter-C input contribution (D) under three water levels at the end of the expe

---

## Author Response (AR1)

Dear Dr Michael Weintraub

Thank you very much for giving us the opportunity to revise our manuscript entitled "Factors controlling *Carex brevicuspis* leaf litter decomposition and its contribution to surface soil organic carbon pool at different water levels". We carefully considered all the comments from the two anonymous reviewers. These comments are very valuable and provide great help for us to revise and improve the clarity and rigour of the presentation of our work. We have read all the comments carefully, responded point-by point, and revised the manuscript accordingly.

In the revised manuscript, revised and corrected contents (including references) are marked in red. We hope the revisions makes our manuscript more worthy of publication. Our detailed responses to the comments are as follows:

**Response to Reviewer #1**

Comment 1.1

General comments

Zhu et al. not only identified the major factor controlling leaf litter decomposition as water level, but also revealed its working approach in natural freshwater wetlands. The systematic and scientifically sound design delivered new insights into wetland leaf litter decomposition processes and consequences. I recommend to be accepted after revision.

Response 1.1

We appreciate the positive evaluations from the reviewer on our work and are grateful for the reviewer for recognizing the potential impact of our work.

Comment 1.2

Specific comments

Abstract: L25-27: The key rate values should be added.

Response 1.2

Thank you very much for your detailed suggestion. L25-27 has been changed to:

*The percentage litter dry weight loss and the instantaneous litter dry weight decomposition rate were the highest at +25 cm water level (61.8%, 0.01307d$^{-1}$), followed by the 0 cm water level (49.8%, 0.00908 d$^{-1}$), and the lowest at -25 cm water level(32.4%, 0.00527 d$^{-1}$).* See Line 25-27 in the revised manuscript.

Comment 1.3

L33: Change "strengthen" to "increase".

Response 1.3

Changed as suggested, Thank you!

Comment 1.4

L35: Change "influences" to "influenced".

Response 1.4

Changed as suggested, Thank you!

Comment 1.5

L36: Change "affects" to "and affected".

Response 1.5

Changed as suggested, Thank you!

Comment 1.6

Introduction: L40: Change "25" to "25%".

Response 1.6

Thank you for your suggestion, but after all things considered, we deleted the sentence.

Comment 1.7

L66-69: Move to M & M.

Response 1.7

Thank you for the suggestion. We have moved *"Dongting Lake (28°30'–30°20'N, 111°40'–113°10'E) is the second largest freshwater lake in China. It is connected to the Yangtze River via tributaries. Dongting Lake wetlands are characterized by large seasonal fluctuations in water level (≤ 15 m) and are completely flooded during June–October and exposed during November–May (Chen et al., 2016)."* to "Materials and methods" section as suggested. Please see L83-L86 in the revised manuscript.

Comment 1.8

L71: Species is not a vegetation

Response 1.8

Thank you for the reminder. This sentence has been rephrased as "*Carex brevicuspis* is a dominant species in the Dongting Lake wetland". Please see L69 in the revised manuscript.

Comment 1.9

L82: Unclear. "decomposition controls differs"?

Response 1.9

We are sorry for the ambiguity. It means that the intrinsic control factors are different at different water levels. This sentence has been rephrased as "*the intrinsic factors that control litter decomposition rate at three water levels are different*" to avoid confusing. Please see L78-79.

Response 1.10

Thank you for the comment. This sentence has been moved to line 88-89 in the revised manuscript as suggested.

Response 1.11

We are very sorry for our negligence of detailed description about the belowground water. The belowground water is extracted from the well in the experiment site by the water pump. We have added this information in M&M part. Please see L98-100 in the revised manuscript.

Comment 1.12

L105: How to arrange the 15 litterbags (10 cm × 15 cm) within each soil cores (40 cm diameter)?

Response 1.12

We are very sorry for our negligence. Litter bags were laid flat on the surface of the soil. Each litter bag was not filled, and there are a little overlap between the litter bags where there is no litter. Please see L104-105 in the revised manuscript.

Comment 1.13

L170: Multiple regression method should be added.

Response 1.13

Thank you for the suggestion. We have added the following sentence in the statistical analyses section.

*The intrinsic litter decomposition rate-limiting factors were analyzed by stepwise regression method in a multiple regression model.* Please see L178-179 in the revised manuscript.

Comment 1.14

L198-201, Table 1: Why not choose the same variables in every regression model? Please explain or give the methodology basis.

Response 1.14

We are very sorry for our negligence. Stepwise regression is used to calculate the regression model, the variables were the result of Stepwise regression model filtering, so the variables were different. The regression model methodology has been added in section 2.6, in L178-179.

Comment 1.15

Figure 1: The full words of S, L and D should be added in the caption.

Response 1.15

Thank you for the detailed suggestion. The following paragraph has been added in the Figure 1 caption.

*L represents litter which was distributed on the soil surface in 15 litter bags to observe the effects of leaf litter input on soil carbon pool; S represents soil which was designated the litter removal control; D represents decomposition which was distributed on the soil surface in 15 litter bags to monitor the litter decomposition rate and process.* Please see L437-441 in the revised manuscript.

Comment 1.16

L227: K-value should be kept consistent with k occurred in M & M.

Response 1.16

We are very sorry for our negligence. K-value in L227 in the original manuscript has been changed to the instantaneous litter dry mass decay rate ($k$) that was kept consistent with k occurred in M & M. Please see L233-235 in the revised manuscript.

Comment 1.17

L230: Please specify which results.

Response 1.17

These results are that the percentage litter dry weight loss and the decomposition rate increased with water level supported our first hypothesis.

"Hence, the percentage litter dry weight loss and the decomposition rate increased with water level. These results supported our first hypothesis." has been rephrased as following: *"Hence, the percentage litter dry weight loss and the decomposition rate increased with water level, which supported our first hypothesis."* Please see L235-237.

Comment 1.18

L232-233, 244-245: Not always the truth. Water will inhibit most decomposition as well for lack of oxygen.

Response 1.18

We are sorry for the ambiguity. The purpose of quoting this sentence is to show that there are existing studies supporting the results in our study, and to provide a scientific and reasonable explanation for my research results. The sentences have been changed to: *Related research showed that the wetland water level strongly affects litter leaching and microbial decomposition (Peltoniemi et al., 2012). Molles et al (1995) also found that compared with the terrestrial environment, in wetland, water promotes litter leaching and microbial metabolism, thereby accelerating litter decomposition. Moreover, water infiltration into litter also increases relative leaching loss (Molles et al., 1995).* Please see L238-242 in the revised manuscript.

Comment 1.19

L251-259: It's more interesting to discuss why the same litter subject to various water levels were mainly controlled by different factor?

Response 1.19

Thank you for the suggestion. This is mainly because that in different water levels, the rates of N lost were different. *At the 0 cm and +25 cm water level, N is rapidly lost and the L/N ratio significantly increases. Thus, L/N is the main internal limiting factor at the 0 cm and +25 cm water level.* Please see L265-267 in the revised manuscript.

Comment 1.20

Response 1.20

References (Gao et al., 2016;Chen et al., 2018) have been added in the text. Please see L290-291.

Response 1.21

We are sorry for the mistake. The conclusion has been rephrased as "*In this study, we quantified the contribution of leaf litter decomposition on soil surface organic carbon pools (S-SOCPs) under different water level conditions. Appropriate flooding (+25 cm water level treatment in our study) can significantly promote the decomposition of litter and contribute about 13.75% organic carbon to S-SOCPs. Under waterlogging condition (0 cm water level), litter decomposition, which mainly controlled by microbial activity, contributed 4.73% organic carbon to S-SOCP. However, under relative drought conditions (-25 cm water level treatment in our study), litter decomposition only contributes about 2.51% organic carbon to S-SOCP, which is largely ascribed to the slower decomposition rate and soil carbon lost by microbe metabolism (i.e., actinomycetes). We also found that lignin or lignin/N content were intrinsic factors controlling the litter decomposition rate in Carex brevicuspis. In Dongting Lake floodplain, the groundwater decline due to climate change and human disturbance would slow down the return rate of organic carbon from leaf litter to the soil, and facilitate the S-SOCP loss.*" Please see L299-310 in the revised manuscript.

Comment 1.22

L291-293: Beyond the support of this study.

Response 1.22

We accept this comment and this part had been deleted. Our conclusion has been rephrased as follows: *In this study, we quantified the contribution of leaf litter decomposition on soil surface organic carbon pools (S-SOCPs) under different water level conditions. Appropriate flooding (+25 cm water level treatment in our study) can significantly promote the decomposition of litter and contribute about 13.75% organic carbon to S-SOCPs. Under waterlogging condition (0 cm water level), litter decomposition, which mainly controlled by microbial activity, contributed 4.73% organic carbon to S-SOCP. However, under relative drought conditions (-25 cm water level treatment in our study), litter decomposition only contributes about 2.51% organic carbon to S-SOCP, which is largely ascribed to the slower decomposition rate and soil carbon lost by microbe metabolism (i.e., actinomycetes). We also found that lignin or lignin/N content were intrinsic factors controlling the litter decomposition rate in Carex brevicuspis. In Dongting Lake floodplain, the groundwater decline due to climate change and human disturbance would slow down the return rate of organic carbon from leaf litter to the soil, and facilitate the S-SOCP loss.* Please see L299-310 in the revised manuscript.

The references in the responses were listed as follows:

Chen, H. Y., Zou, J. Y., Cui, J., Nie, M., and Fang, C. M.: Wetland drying increases the temperature sensitivity of soil respiration, Soil Biology & Biochemistry, 120, 24-

27, 10.1016/j.soilbio.2018.01.035, 2018.

Gao, J. Q., Feng, J., Zhang, X. W., Yu, F. H., Xu, X. L., and Kuzyakov, Y.: Drying-rewetting cycles alter carbon and nitrogen mineralization in litter-amended alpine wetland soil, Catena, 145, 285-290, 10.1016/j.catena.2016.06.026, 2016.

**Response to Reviewer #2**

Comment 2.1

Line 20 and 46: I recommend including (Cao et al. 2020), and references therein, that address aboveground litter decomposition on SOC pools.

Response 2.1

Thank you for your recommendation, we had cited (Bowden et al., 2014; Cao et al., 2020) in the revised manuscript in L45-46.

Comment 2.2

Line 30: The SOC increase due to litter application (Figure 5d) appears to be calculated from Figure 5a, but I could not reconcile the value for -25 cm. While Figure 5 does seem to support that litter increases SOC, I have two concerns about this presentation. First, the differences in Figure 5a for the 0 cm water levels is labelled as significant, but the error bars clearly overlap. Please clarify. Second, and a potential fundamental flaw in the data presentation, is that the baseline SOC is not provided and there is no way to know if SOC changed throughout the course of the experiment. It appears based on Equation 4 you did measure baseline SOC for each core (_18 g kg-1)? Please clarify. In either case, a significant observation, which is not discussed, is that the no-litter treatments resulted in very large SOC increases and adding litter resulted in a small additional increase.

Response 2.2

For the first concern, we are sorry for the mistake of the error bars. Figure 5a has been reconstructed.

For the second concern, the soil cores were collected from the same site, and the baseline SOC was 63.32g kg$^{-1}$. The aim of this study was to clarify the impact of litter addition on SOC, so we did not present the SOC baseline. To avoid confusion, we have added the baseline SOC (63.32g kg$^{-1}$) in section 2.2 in L109-110 in the revised manuscript.

We are sorry for the negligence. The SOC differences among three water levels were caused by different soil mineralization in different environments. Soil mineralization in aerobic environment (-25 cm) was significantly higher than that in the flooded environment (0 cm, +25 cm) (Qiu et al., 2018), so the SOC at -25 cm water level was lower than the other two water levels. We had added the sentences "*In wetlands, water level fluctuations could readily cause carbon loss (Gao et al., 2016; Chen et al., 2018). The SOC differences among three water levels were caused by different soil mineralization in different environments. Soil mineralization in aerobic environment (-25 cm) was significantly higher than that in the flooded environment (0 cm, +25 cm) (Qiu et al., 2018), so the SOC at -25 cm water level was lower than the other two water levels*" in the revised manuscript in L290-294.

Comment 2.3

Line 40: References for this statement are inappropriate, or incorrectly cited. Means et al. 2016 does not discuss global carbon pools. Whiting and Chanton (2001) is an accurate source for the value you used for wetland carbon stocks, but they cite Schlesinger 1991 as their source, and there are more up-to-date carbon stock estimates, such as (Köchy, Hiederer, and Freibauer 2015). Cao et al., 2017, is a secondary reference, like Whiting and Chanton (2001), and neither reflect the range of wetland soil carbon (25– 63%) you provide. The value in Whiting and Chanton (2001) is 3 – 68% (secondary references) and the value in Cao et al. (2017) is 12 – 15% (also a secondary reference). Use of the most up-to-date sources and an accurate reflection of those sources adds value to the manuscript. I recommend adding a citation such as (Kayranli et al. 2010), which could also be useful in your discussion considering what happens to the SOC after it is leached from the litter into the soil.

Response 2.3

We are sorry for the mistake and thank the reviewer very much for the commendation and suggestion. We have add the value into the manuscript, and cited the references (Kayranli et al., 2010;Kochy et al., 2015). The sentences have been rephrased as follows:

*Wetlands are important terrestrial carbon pools. They contain between 82 and 158 Pg SOC, which depending on the definition of "wetland" (Kayranli et al., 2010; Kochy et al., 2015).* Please see L40-41 in the revised manuscript.

Comment 2.4

Line 52: The Sun et al., 2019 study does not support the statement that litter decomposition stabilized the soil organic carbon pool. Litter decomposition made

DOC more mobile and labile, which the authors suggested could lead to SOC stability after processing by soil microbes.

Response 2.4

We are sorry for the mistake. The sentences have been rephrased as: *In contrast, a recent study found that litter decomposition stabilized the soil carbon pool after processing by soil microbes in the Jiaozhou Bay wetland (Sun et al., 2019).* Please see L51-52 in the revised manuscript.

Comment 2.5

Lines 54-56: Aerts (1997) addresses litter decomposition in non-wetland sites where shredder invertebrates (detritovores) are important, but their role in wetland settings is more uncertain (Inkley, Wissinger, and Baros 2008). Shredding would be an important physio-chemical control on DOC leaching.

Response 2.5

We thank the reviewer very much for the commendation and suggestion.    The references have been changed, and the sentences have been rephrased as follows:

*Litter decomposition is a physicochemical processes that reduces litter to its elemental chemical constituents (Berg and McClaugherty, 2003). Litter decomposition rates are determined mainly by environmental factors (climatic and soil conditions), litter quality (litter composition such as C, N, and lignin content) and decomposer organisms (microorganisms and invertebrates) (Yu et al., 2020;Yan et al., 2018).* Please see L53-57.

Comment 2.6

Line 62: Zhang 2019 supports the statement that water levels affected microbial activity, but leaching and fragmentation were only discussed, not measured.

Response 2.6

We are sorry for the ambiguity. The references have been changed to (Van de Moortel et al., 2012), which designed a leaching experiment to clarify the leaching process of litter decomposition. Please see L60-62 in the revised manuscript.

Comment 2.7

Line 64: This is a mischaracterization of the Upton, 2018 study. Perhaps a better reference is (Hoyos-Santillan et al. 2015). However, clarification is needed because

Hoyos-Santillan states that roots (not litter) are the main source of SOC in peatlands, but litter strongly influences root decomposition rates, particularly near the surface.

Response 2.7

We are sorry for our carelessness. The sentence has been rephrased as follows: *Leaf litter contributes more to soil organic carbon than fine roots (Cao et al., 2020), litter also strongly influences root decomposition rates, particularly near the soil surface (Hoyos-Santillan et al., 2015).* Please see Line 64-66.

Comment 2.8

Line 151. The Olson (1963) simple decay model assumes constant k, which you demonstrated is not a constant. Although use of this decay model is common in the literature, it is an oversimplification. This does not adversely affect your comparative analysis, but the paper would be strengthened with a more sophisticated analysis, such as a double exponential decay model (Berg 2014 or Wider and Lang 1982).

Response 2.8

Thank you for the constructive suggestions. We have modified the model based on your suggestion to highlight the instantaneous rate variation of litter decomposition. The model is:

$$M_{t_n} = M_{t_{n-1}} e^{-k_n(t_n - t_{n-1})}$$

Where $M_{t_n}$ is the litter dry matter weight at $n$th sampling (g), $M_{t_{n-1}}$ is the litter dry matter weight at (n-1)th sampling (g), $t_n - t_{n-1}$ is the time between the $n$th and (n-1)th sampling, $k_n$ is the instantaneous decomposition rate at the $n$th sampling. (Please see line 154-159 in the revised manuscript.

This model would be more accurate. The result was as following: The instantaneous litter decomposition rate was highest at initial and slowly decreased and stabilized at all three water levels. The maximum decomposition rates at the -25 cm, 0 cm, and +25 cm water levels were 0.00527 d⁻¹, 0.00908 d⁻¹, and 0.01307 d⁻¹, respectively (Fig. 2b). Please see L192-196 in the revised manuscript.

[Figure]

Based on the change of the instantaneous decomposition rate, we recalculated the multiple regression model which was used to analyze the intrinsic litter decomposition rate-limiting factor. The models are as follows (Table 1):

| Water level (cm) | Multiple regression model | $F$ | $R^2$ | $P$ |
|---|---|---|---|---|
| -25 | $R = -\mathbf{0.715L} - 0.443C + 0.033$ | 5.738 | 0.727 | 0.006 |
| 0 | $R = -\mathbf{928LN} - 0.233CN + 0.023$ | 5.928 | 0.927 | $< 0.001$ |
| +25 | $R = -\mathbf{0.717LN} + 0.016$ | 9.543 | 0.793 | 0.002 |

The multiple regression model of the instantaneous litter decomposition rate and the litter properties showed that at the -25 cm, the main decomposition rate-limiting factor was the lignin concentration, whilst at 0 cm and +25 cm water level, the main litter decomposition rate-limiting factor was the lignin/N ratio. Please see L204-207 in the revised manuscript.

Comment 2.9

Line 266: Doesn't your argument imply C. brevicuspis, due to it's lower lignin content, would return less carbon to the soil compared to the other plants you cited? The manuscript may be strengthened, and have a wider inference, if you listed and compared decomposition rates and lignin contents of wetland plants including C. brevicuspis.

Response 2.9

Thank you for the suggestion. But we just intended to clarify that the lignin content of *C. brevicuspis* leaf litter was lower than the other plants, so the *C. brevicuspis* leaf litter was more easily to be leached and then contributed more to the SOC pool. Due to the different environment, the litter decomposition rates were different, so we didn't compared decomposition rates. On the other hand, the aim of our study was to explore the contribution of litter decomposition to SOC pool, instead of the relationship between lignin content and litter decomposition rate. Taking all these into account, we didn't list and compare decomposition rates and lignin contents of wetland plants.

Comment 2.10

Line 284: Your conclusion contains a significant amount of new and largely unsupported discussion material. Conclusions should stick to what you were able to show in your experiments.

Response 2.10

Thank you very much for the comment. The conclusion part has been rephrased as follows: *In this study, we quantified the contribution of leaf litter decomposition on soil surface organic carbon pools (S-SOCPs) under different water level conditions. Appropriate flooding (+25 cm water level treatment in our study) can significantly promote the decomposition of litters and contributed about 16.93% organic carbon to S-SOCPs. Under waterlogging condition (0 cm water level), litter decomposition, which mainly controlled by microbial activity, contributed 9.44% organic carbon to S-SOCP. However, under relatively drought condition (-25 cm water level treatment*

*in our study), litter decomposition only contribute about 2.51% organic carbon to S-SOCP, which is largely ascribe to the slower decomposition rate and soil carbon lost by metabolism of the microbes (i.e. actinomycete). We also found that lignin and/or lignin/N content were intrinsic factors controlling litter decomposition rate in Carex brevicuspis. In Dongting Lake floodplain, the groundwater decline which was caused by the climate change and human disturbance would slowdown the return rate of organic carbon from leaf litter to soil, and facilitate the S-SOCP loss.* Please see L299-310 in the revised manuscript. We hope this modification can meet the requirements.

Comment 2.11

Table 1: Is the lignin content (L) in the regression model the initial lignin content?

Response 2.11

Sorry we didn't clearly define the model indictors. The regression model is used to analyse the intrinsic litter decomposition rate-limiting factor. We added "All indicators used to analyse the model was refered to the content at each time point" in L432-433 in the revised manuscript.

Comment 2.12

Figure 3d (LRRI%) is nearly identical to Figure 2a (litter dry weight loss, %), which seems counter-intuitive unless lignin were the sole material being mineralized during the decomposition process. Did you measure lignin content at each time point?

Response 2.12

We measured lignin content at each time point. In fact, the results of stepwise regression analysis showed that lignin content is the main intrinsic litter decomposition rate-limiting factor, which is consistent with the figure 3d.

Technical corrections:

Comment 2.13

Line 23: Carex brevicuspis may be ubiquitous to wetlands in China; however, is this true globally?

Response 2.13

We are sorry for the ambiguity. This sentence has been rephrased as: The *Carex* genus is ubiquitous to global freshwater wetlands. Please see L23 in the revised manuscript.

Comment 2.14

Line 25: Is "mass loss" = "carbon release"? If so, one of these phrases is redundant.

Response 2.14

Thank you for remind us. In our opinion, mass loss includes not only carbon release but also other elements release, such as N, P. but because of the high proportion of carbon release, the trend of mass loss and carbon release are similar. Mass loss reflected the whole process of litter decomposition, while carbon release reflected the process of carbon release.

Comment 2.15

Line 82: The way you stated your hypotheses imply you have tested causal factors, which you did not. Specifically, leaching, fragmentation and infiltration.

Response 2.15

Thank you for reminding us. We have rephrased the hypotheses as follows: First, water level has a significant effect on litter decomposition. Second, the intrinsic limiting factors may be different among three water levels. Third, the contribution of leaf decomposition to S-SOCP was relatively higher at the +25 cm water level. Please see L77-80 in the revised manuscript.

Comment 2.16

Line 102: Did the litter bags float? Did you need to pin them in contact with the soil surface?

Response 2.16

We are sorry for the negligence. All litter bags were fixed to the soil surface with bamboo sticks. And the sentence has been added in section 2.2 in L107-108.

Comment 2.17

Line 105 - 107: Clarify how many soil cores were used in each pond for each purpose and how they were prepared (e.g. were soils blended prior to starting the experiment). The text is confusing.

Response 2.17

We have clarified that all the soil cores were undisturbed soil. The experiment was conducted in nine cement ponds (2 m × 2 m × 1 m) (please see L107-108 in revised manuscript). Three soil core sets were placed in each pond. One was designated the litter removal control (S), the second was distributed on the soil surface in 15 litter bags to observe the effects of leaf litter input on soil carbon pool (L), and the third was distributed on the soil surface in 15 litter bags to monitor the litter decomposition rate and process (D) in L102-108 in revised manuscript.

Comment 2.18

Line 183: You use capital letters (Fig 2A) in text references, but lower-case letters in the Figures.

Response 2.18

We are sorry for the mistake. The capital letters in the text references have been changed into lower-case letters.

Comment 2.19

Line 187/188 and Line 247: I would not interpret your data that decomposition rates "rapidly increased" – the decomposition rate at time t=0 is undefined.

Response 2.19

We are sorry for our obscure writing. The sentences have been rephrased as: *The instantaneous litter dry weight decomposition rate was highest at initial and slowly decreased and stabilized at all three water levels*. Please see L193-194 in revised manuscript.

Comment 2.20

Figure 4: Are these figures reporting mass? The units are nmol g-1.

Response 2.20

We are sorry for that we didn't clearly define the calculation method about microbial community structure. These figures were used to report the PLFA molar mass concentration. This is a common way to show the microbial content (Zhao et al., 2015). We calculated PLFA mass content first, PLFA (ng g$^{-1}$ dry soil) = (Response of

PLFA/Response of 19:0 internal standard) $\times$ concentration of 19:0 internal standard $\times$ (volume of sample / mass of soil). Concentration of 19:0 internal standard: 5 $\mu g\ ml^{-1}$, volume of sample: 200µl,mass of soil: 8g dry soil. And then we calculated PLFA molar mass concentration, PLFA (n mol $g^{-1}$ dry soil) = PLFA (ng $g^{-1}$ dry soil)/ relative molecular mass. Please see L143-147 in revised manuscript.

The references in the responses were listed as follows:

Cao, J. B., He, X. X., Chen, Y. Q., Chen, Y. P., Zhang, Y. J., Yu, S. Q., Zhou, L. X., Liu, Z. F., Zhang, C. L., and Fu, S. L.: Leaf litter contributes more to soil organic carbon than fine roots in two 10-year-old subtropical plantations, Science of the Total Environment, 704, 8, 10.1016/j.scitotenv.2019.135341, 2020.

Hoyos-Santillan, J., Lomax, B. H., Large, D., Turner, B. L., Boom, A., Lopez, O. R., and Sjogersten, S.: Getting to the root of the problem: litter decomposition and peat formation in lowland Neotropical peatlands, Biogeochemistry, 126, 115-129, 10.1007/s10533-015-0147-7, 2015.

Kayranli, B., Scholz, M., Mustafa, A., and Hedmark, A.: Carbon Storage and Fluxes within Freshwater Wetlands: a Critical Review, Wetlands, 30, 111-124, 10.1007/s13157-009-0003-4, 2010.

Kochy, M., Hiederer, R., and Freibauer, A.: Global distribution of soil organic carbon - Part 1: Masses and frequency distributions of SOC stocks for the tropics, permafrost regions, wetlands, and the world, Soil, 1, 351-365, 10.5194/soil-1-351-2015, 2015.

Qiu, H. S., Ge, T. D., Liu, J. Y., Chen, X. B., Hu, Y. J., Wu, J. S., Su, Y. R., and Kuzyakov, Y.: Effects of biotic and abiotic factors on soil organic matter mineralization: Experiments and structural modeling analysis, Eur. J. Soil Biol., 84, 27-34, 10.1016/j.ejsobi.2017.12.003, 2018.

Yan, J. F., Wang, L., Hu, Y., Tsang, Y. F., Zhang, Y. N., Wu, J. H., Fu, X. H., and Sun, Y.: Plant litter composition selects different soil microbial structures and in turn drives different litter decomposition pattern and soil carbon sequestration capability, Geoderma, 319, 194-203, 10.1016/j.geoderma.2018.01.009, 2018.

Yu, X. F., Ding, S. S., Lin, Q. X., Wang, G. P., Wang, C. L., Zheng, S. J., and Zou, Y. C.: Wetland plant litter decomposition occurring during the freeze season under disparate flooded conditions, Science of the Total Environment, 706, 9, 10.1016/j.scitotenv.2019.136091, 2020.

Zhao, J., Zeng, Z. X., He, X. Y., Chen, H. S., and Wang, K. L.: Effects of monoculture and mixed culture of grass and legume forage species on soil microbial community structure under different levels of nitrogen fertilization, Eur. J. Soil Biol., 68, 61-68, 10.1016/j.ejsobi.2015.03.008, 2015.

Again, we greatly appreciate the editor and reviewers for all the insightful comments. We worked hard to be responsive to them. We sincere thank the editor and reviewers for taking the time and energy to help us improve the manuscript. We look forward to hearing from you.

Sincerely yours

Lianlian Zhu, on behalf of co-authors

E-mail: zhulianlian426@163.com

[revised manuscript text omitted]

---

## Author Response (AR2)

Dear Dr Michael Weintraub

Thank you and all of the reviewers for the positive comments on our work. We have revised the manuscript accordingly. In the revised manuscript, revised and corrected text (including references) are marked in red. We hope the revisions makes our manuscript more worthy of publication. Our detailed responses are as follows:

**Response to Reviewers**

Comment 1

L31: Change "application" to "addition".

Response 1

Changed as suggested, thank you.

Comment 2

L66: Change "SOC pool" to "S-SOCP".

Response 2

Corrected as suggested, Thank you!

Comment 3

L74: Change "litter" to "leaf litter".

Response 3

Changed as suggested, thank you very much.

Comment 4

L131: Not "TOC analyser" but "measured with a TOC analyser"

Response 4

Changed as suggested, Thank you!

Comment 5

L 202, 287: Change "-25-cm" to "25 cm".

Response 5

We are sorry for the carelessness, and changed it as suggested, Thank you!

Comment 6

L257: Change "onset" to "initial".

Response 6

Revised as suggested, thank you.

Comment 7

Fig. 2b, Fig. 4 and Fig. 5: The error bar should be bolder.

Response 7

The figures had been adjusted as suggested, Thank you very much!

Comment 8

L376-377, L378-380:Please check the reference format.

Response 8

We are sorry for the negligence, and the references were revised in accord with the journal style.

Again, we appreciate the editor and reviewers for taking the time and energy to help us improve the manuscript. We look forward to hearing from you.

Sincerely yours

Lianlian Zhu, on behalf of co-authors

E-mail: zhulianlian426@163.com

[revised manuscript text omitted]